# A Biocultural Study on Gaoligongshan Pig (*Sus scrofa domesticus*), an Important Hog Landrace, in Nujiang Prefecture of China

**DOI:** 10.3390/biology11111603

**Published:** 2022-11-02

**Authors:** Yanan Chu, Chen Lin, Zhuo Cheng, Xingcen Zhao, Yanxiao Fan, Binsheng Luo, Chunlin Long

**Affiliations:** 1Key Laboratory of Ecology and Environment in Minority Areas (Minzu University of China), National Ethnic Affairs Commission, Beijing 100081, China; 2College of Life and Environmental Sciences, Minzu University of China, Beijing 100081, China; 3College of Education, Yunnan University of Business Management, Kunming 650106, China; 4Lushan Botanical Garden, Jiangxi Province and Chinese Academy of Sciences, Jiujiang 332001, China; 5Institute of National Security Studies, Minzu University of China, Beijing 100081, China

**Keywords:** biocultural diversity, Gaoligongshan pig, forage plants, traditional knowledge

## Abstract

**Simple Summary:**

The biocultural diversity associated with Gaoligongshan pigs was investigated in the present paper. Six villages in Laowo Township of Lushui City in the Nujiang River watershed, where the Gaoligongshan pigs are primarily raised, were selected to collect information and data related to biocultural diversity. Participatory surveys and semi-structured interviews were conducted to document the semi-wild and free-ranging management pattern of Gaoligongshan pigs. The wild, cultivated, and medicinal plants used in rearing pigs were documented in terms of local knowledge. Most wild forage species are from Polygonaceae and Compositae while Poaceae dominated in cultivated forage plants. The local Bai and Lisu ethnic minorities have accumulated a wealth of biocultural knowledge related to diet, medicine, festivals, and rituals during their long-term rearing activities, and forming a cultural complex about Gaoligongshan pigs. This study demonstrated that the semi-wild and free-ranging management model of the Gaoligongshan pig is consistent with the local natural environment, traditional culture, economic level, and the breed’s characteristics.

**Abstract:**

Over 80% proteins consumed by the local people in Nujiang Prefecture of Southwest China, a remote and mountainous area in the Eastern Himalayas, are from pork, or Gaoligongshan pig (a landrace of *Sus scrofa domestica* Brisson). Previous research on the Gaoligongshan pig has focused on nutritional composition, production performance, and genetic resource characteristics, but neglected the reasons behind the local people’s practice. From 2019 to 2022, we have used ethnobiological research methods to comprehensively document the traditional rearing and management patterns and the traditional culture associated with Gaoligongshan pigs. The results show that Gaoligongshan pigs graze in mixed herds with cattle and sheep during the day and prefer to eat 23 wild plant species, in which 17 species have medicinal values. At night, the pigs are artificially fed and rest in the pigsty. The local Bai and Lisu people have developed a creative food culture, rituals, and festivals culture associated with Gaoligongshan pigs. Overall, the biocultural diversity of Gaoligongshan pig contributes to the in situ conservation of genetic diversity of this important hog landrace, and supports rural development in this remote area.

## 1. Introduction

Animal husbandry has always played a crucial role in the long history of human social development [1,2]. As one of the most significant biological resources of the country, livestock and poultry genetic resources play an important supporting role in agricultural development [3]. For smallholder economies, raising livestock can, on one hand, provide essential nutritional elements for the human body, including meat and milk, to meet daily dietary needs. On the other hand, it can serve as labor, such as plowing and ploughing fields, to improve the efficiency of farm work. In addition, it can also be used as an economic commodity for commercial trading and monetary exchange [4]. Livestock genetic resources should not only provide superior, safe, and diversified livestock products for the market but also contribute to the promotion of farmers’ income, sustainable use of resources, and ecological and cultural heritage protection [3]. It lays a solid material foundation for food security, rural revitalization, biodiversity conservation and ecological environment construction [5,6,7,8]. Genetic resource protection requires a combination of traditional knowledge of indigenous peoples and modern scientific knowledge, closely related to the knowledge about the biocultural diversity of indigenous communities. In summary, combing and documenting the range of biocultural diversity knowledge of indigenous communities about traditional livestock farming is essential to achieve holistic conservation of livestock farming in the context of traditional knowledge, as well as provide a conventional scientific basis for the future development of livestock farming.

China is an internationally renowned livestock industry powerhouse, contributing over 40% to the development of the world’s livestock industry. The genetic resources of livestock and poultry account for about one-tenth of the world’s total resources [7], and the production of livestock and poultry meat accounts for about 29% of the world’s total production, with the number of live pigs produced ranking first in the world. During thousands of years of breeding and domestication, livestock and poultry have gradually developed genetic characteristics of rough feeding resistance, disease resistance, tasty meat quality and stable genetic performance [5]. At the same time, China is also one of the countries in the world where the genetic resources of stored poultry are most seriously threatened. The overall trend of decline in China’s livestock and poultry genetic resources is due to the pursuit of profit, the elimination of local breeds that lack market competitiveness, the impact of the introduction of alien species, the limitations of conservation policies and regional differences in conservation. According to statistics, local livestock and poultry breeds that are endangered and on the verge of extinction account for about 18% of the total number of local breeds in the past 20 years, of which 15 are in danger, 44 are on the verge of extinction and 17 are extinct. This trend will be further intensified with the increase in intensification and a large number of introductions [6]. Therefore, it is imperative to adopt effective management strategies to conserve the precious genetic resources.

The Gaoligongshan pig (GP, *Sus scrofa domestica* Brisson) is a rare landrace native to the Nujiang River watershed in China, reared in a semi-wild and free-ranging mode, living in the forests. GP is highly adaptable, resistant to roughage, has well-developed tendons and is highly resistant to disease [9,10,11]. The high content of fatty acids, linoleic acid, and linolenic acid in the meat of GP is highly effective in preventing and treating diabetes and cardiovascular diseases in humans [12,13,14,15]. GP is a high-quality pig breed with healthy performance and has been included in *Animal Genetic Resources in China* [10]. However, due to the biological characteristics of GP, such as slow growth rate, weight gain generally remains within 37 to 38 kg over a year; small size, the weight of adult pigs is basically stable at around 58 kg; and the farrowing rate is significantly lower than that of other pig breeds. Because of its higher selling price than that of other pig breeds, the GPs’ market capacity is also limited. All of these reasons led to a drastic reduction in their population scale [10]. Their genetic resources are in danger of becoming endangered [16]. Previous studies on GP have primarily focused on genetic analysis and metabolomics at the genetic level [17], with less attention paid to systematic conservation studies incorporating the biocultural diversity dimension [18]. This study attempts to the perspective of the biocultural diversity approach to (1) investigate the traditional rearing and management experiences of GP reared by different ethnic groups such as the Lisu and Bai of Nujiang; and (2) reveal the biocultural significance of GP in local people’s diet, medicine, religion, and rituals, while pointing out the current difficulties faced, and proposing suggestions for future development.

## 2. Materials and Methods

### 2.1. Study Area

The research area of this study covers Gaoligongshan Mountains and surrounding areas, mostly the Nujiang Lisu Autonomous Prefecture in West Yunnan. Located in the biodiversity hotspot of the Eastern Himalayas, there are rich biological resources, history and culture, and diverse folks and customs in the Gaoligongshan Mountains. Conservation of biodiversity and boosting sustainable development of local communities are the main objectives of this biosphere reserve. People in Nujiang Prefecture and surrounding areas still cultivate and rear a lot of old varieties of crops and domestic animals including Gaoligongshan pig (GP). Lushui of Nujiang Prefecture is the main production zone of GPs. It is located between two mountains, the Gaoligongshan Mountains in the west and the Bilo Snow Mountains in the east, with the Nujiang River running from north to south. The maximum altitude of Lushui reaches 3606 m above sea level and its minimum altitude is 1080 m. The climate is a semi-mountainous, vertically distributed wet zone, with an annual average temperature of 14–16 °C, an annual average of 2500 h of sunshine, and rainfall concentrated between July and September, which accounts for half of the annual rainfall. Laowo, a township with famous GP breeding and GP products, was selected for intensively investigating and collecting data and information about GP. The main linguistic groups breeding GP in the prefecture are Lisu, Bai, Han, and Yi.

### 2.2. Data Collection

From 2019 to 2022, ethnobiological investigations were conducted in six villages, namely Zhongyuan, Laowo, Ronghua, Yinpo, Yunxi, and Chongren in Laowo Town, Lushui City, where most Gaoligongshan pigs are being raised (Figure 1). Before the investigations, it was ensured that all respondents understood the purpose of the survey and were willing to participate in it. The data collection methods included semi-structured interviews and participatory surveys [19,20]. A snowball sampling method was used to conduct semi-structured interviews with 79 key informants from the Bai, Lisu, and Han linguistic groups. The Bai population is account for 46%, with a total of 37 people; the Lisu population is 14%, with a total of 11 people; and the Han population is 40%, with a total of 31 people. Informants included 21 females and 68 males, with an average age of 51.6 years, from 10 to 84 years old, who had come into close contact with this pig farming during their lifetime and had practiced pig farming for an average duration of nearly 24.5 years. They are mostly farmers, together with small proportion of pork product processors, agrarian shop owners, and veterinarians. Their education level was mainly primary and junior high school, accounting for 80% of the total. The interview time with each informant was more than 30 min.

The semi-structured interview revolves around questions see Table 1. Participatory surveys were used for grazing with local farmers, tracking and recording the grazing time, activity range, and feeding preferences of Gaoligongshan pigs. The specimens of forage plants were collected (Figure 2). The nomenclature of all vascular plants followed *Flora of China* [21], and World Flora Online “www.worldfloraonline.org (accessed on 9 september 2022)” as well. The voucher specimens were deposited in the Herbarium of the College of Life and Environmental Sciences, Minzu University of China, in Beijing.

### 2.3. Data Analysis

Information from interviews with 79 key informants was collated to calculate the feeding preferences of Gaoligongshan pig (GP) based on the frequency of occurrence of plant species mentioned by the informants. Their F-values (frequency of occurrence) were calculated according to the following formula. It was denoted by * if F-value varied between 0% and 20%, or indicated by ** if 20–40%, etc. It would be marked with four stars (*****) if above 80%.
$$F=\frac{N_{m}}{Ni}$$

F-values represent the frequency of plant use; *N_m_* is the number of informants mentioning the plant; *N_i_* is the total number of informants; the more significant the F-values, the more frequently the plant is used. Therefore, the more the GP prefers the plant [22].

Regarding the abundance of forage plants in the field, we assessed the frequency of plant occurrence in 1 m × 1 m sample squares in five different orientations. If the plant was present in only one sample, the abundance was recorded as †; if it was present in two samples, it was marked with ††, etc. It was present in all five samples; the abundance was recorded as ††††. Feed proportion is based on data recorded from key person interviews. 

## 3. Results and Discussions

### 3.1. Traditional Feeding and Management Culture of Gaoligongshan Pigs

#### 3.1.1. Morphological Characteristics of the Gaoligongshan Pigs

The Gaoligongshan pigs are known locally as a native hog landrace that is medium to small in size. The entire coat is predominantly solid black, with six white spots (white on the four hooves, on the forehead top, and on the tail tip), and some long and dense bristles extending to the shoulders (some medium or short). The mouth barrel is pointed and straight, with two or three transverse lines on part, face slightly concave. The belly is large and a little droopy. Its limbs are short and small. The hoof is firm. Its body is compact and firm, partly with some sloping coccyxes [10,23] (Figure 3).

#### 3.1.2. Knowledge of Forage Plants Associated with Gaoligongshan Pigs

Gaoligongshan pig (GP) grow in the mountainous Nujiang River watershed, grazing in mixed herds with cattle and sheep during the day and foraging for wild forage such as fresh plant stems and leaves and grassroots [24,25]. The participatory survey revealed that GP prefers to eat 23 species of wild plants (Table 2). They are from ten families with Asteraceae and Polygonaceae dominated, accounting for five and six species, respectively. Nearly half of plant species have a sour taste, with up to four of the six Polygonum species having a sour taste which indicated that GP likes to eat plants with a sour taste. Asteraceae are also the primary choice of forage plants influenced by human dietary needs and medicinal value. The feeding parts of GP include the above-ground parts, leaves, and fruits, of which the above-ground parts are the main ones, accounting for 78%. Among these 23 species, the local people also used 17 wild plant species to treat human ailments, focusing on clearing heat and detoxifying the body, reducing swelling and dissipating knots, and appetizing the stomach to stop diarrhea. They have sound therapeutic effects against the common diseases of GP.

During the special stages of growth of GP, such as the little piglet period and the pregnant lactation period, cultivated crops need to supplement nutrients appropriately. Regarding supplementary feeding, from 0–2 months old, GP is mainly breastfed with a small amount of maize flour and rice soup to supplement their nutrition. At 4–5 months of age, the pigs are gradually weaned, and a small amount of forage is added. Feeding is mainly in the form of maize bran mixed with grass 3–4 times a day. After half a year, as the age of the pig increases, the proportion of forage increases, and maize bran is replaced with maize kernels as the main supplementary feeding ingredient, with a stable feeding frequency of three times a day. After 10 months, the diet of GP tends to stabilize, with wild forage dominating during the day and supplemented by cultivated crops such as maize at night (Figure 4). The details of commonly cultivated crops at home are listed in Table 3.

The replenishment of artificially cultivated plants, in addition to meeting the physiological needs of Gaoligongshan pig (GP) at different stages of growth, is also closely related to the changes in seasonal weather [26]. Wild plants generally enter a period of decline at the end of autumn. When the amount of food resources available for GP decreases significantly, farmers will gradually increase the proportion of cultivated plants as foreage. Part-cultivated plants help GPs’ survival in the period of food poverty and partly to fatten them up for the upcoming Chinese New Year slaughter ceremony. It is important to stress that supplementary feeding is not the feeding of artificial feeds. Local people have told us that GPs can suffer from severe gastrointestinal rejection, diarrhea, dehydration, and even death due to artificial feeds. Therefore, for over a hundred years, GPs have adhered to a feeding structure based on forage plants, supplemented by cultivated plants. Farmers have gradually developed a more perfect, semi-wild free-range model to match GP.

#### 3.1.3. The Semi-Wild, Free-Range Model of Gaoligongshan Pig

The Gaoligongshan pig (GP) is raised in a semi-wild and free-range state and is active year-round in dense forests at an altitude of around 2300 m above sea level. This breeding method has greatly increased the average daily movement of the GPs. The GP’s fast-running speed, wide range of movement, and ability to recognize the voice of its owner make it difficult to be hunted by humans and other animals in its natural habit. The fact that it forages in the vast mountain forests contributes to its well-developed tendons, firm meat, and vigorous vitality.

The household rearing rate of GP exceeds 70%, with an average rearing capacity of more than five pigs. There are also large ecological breeding farms built in the Chongren Village. To facilitate the management, people usually build pig pens with wood from the mountains in relatively flat and open locations on the way from villages to the mountains to provide shelter for the GPs, especially during lactation (Figure 5). The bottom of the pens is usually lined with a certain thickness of hay to keep the pigsty dry and avoid inflammation of skin and hoof nails due to prolonged submersion in wet conditions. Typical plants are used for gaskets can be seen in Table 4, with pine needles being the primary material used in winter.

The Gaoligongshan pig’s puberty is usually reached at around four months of age. The period of oestrus and gestation will be influenced by the local climate and the abundance of vegetation [27,28]. The oestrus is sensitive to scent recognition and other perceptions, as well as agitated grunting, increased secretions and climbing across each other, thus attracting male pigs and wild boars within a 3–5 km range for mating to take place [27]. The pregnancy cycle is 142 days, with most farrowing concentrated in June to July. The number of litters born in the first farrowing is about 4–6, and the next litter increases to 7–8. There is no manual intervention for the entire pregnancy and delivery process. The high coverage of forest vegetation and rich biological resources of the Nujiang River watershed provides the possibility of mating with wild boar. The piglets produced after mating are robust, vigorous, disease resistant, and preserve the biological advantages of wild boars. The unique genetic advantage, semi-wild and free-range rearing model, and the preference of taking food from the same source as medicine contribute to the excellent quality of the meat [29].

The survey revealed that the rearing rate of Gaoligongshan pigs is high in the remote villages and is the main source of protein for the local Bai and Lisu people. However, they are generally family-based, still at the stage of a smallholder economy, resulting in the market share being seriously crowded out by large-scale farming [30]. Although the local government will give farmers a certain amount of insurance subsidy, farmers are weak in risk awareness. They only buy insurance for a few breeding pigs. Still, epidemics are more of a mass phenomenon, resulting in the insurance payout not being enough to cover the farmers’ initial investment. In addition, the rising price of supplementary feedstuffs such as *Zea mays* L. has further reduced farmers’ profit margins, making them less able to take the risks of large-scale rearing and struggling to make ends meet [31]. The phenomenon mentioned above has reduced the incentive of farmers to raise Gaoligongshan pigs. Even though, epidemics have always been the biggest obstacle to the development of pig farming [32]. In recent years, the collective extinction of Gaoligongshan pigs caused by African swine fever has been the biggest farming threat to the entire Gaoligongshan pig industry. The disease is highly contagious, spreads quickly, and has a high mortality rate. There are a lack of reliable means of prevention and detection before the onset of the disease, and there are fewer effective drugs to cure the disease after contracting it, causing huge economic losses. Most key informants reported that in Zhongyuan Village, the household rearing rate of Gaoligongshan pig was primarily maintained at 90–95% ten years ago and reduced to 70–75% in 2022, a situation that was also prevalent in the other villages we surveyed. Our research has documented and demonstrated that expanding the feeding and exercise range of the Gaoligongshan pigs through an ecological farming management model that allows them to live in a vast, comfortable natural environment helps them to enhance their immunity and resistance to disease.

#### 3.1.4. The Role of Traditional Chinese Veterinary Medicine in the Prevention and Treatment of Diseases

Surveys have found that Gaoligongshan pig are susceptible to inbreeding-type infectious diseases. Common diseases include contagious blister, epidemic encephalitis, viral gastroenteritis, kidney infection, and dysentery. Local people continue the healing philosophy of traditional Chinese medicine [33], applying medication from indigenous plants and treating diseases (Table 5).

The blistering infectious disease morbidity rate is as high as 70–80%. Symptoms manifest as inflammation, drooling, foot lameness, and fever up to 40.8 °C. Local veterinarians use a recipe called *Wu Huang San* for treating this ailment. *Wu Huang San* is a decoction of a mixture of five herbs, namely *Phellodendron chinense* Schneid., *Coptis teeta* Wall., *Scutellaria baicalensis* Georgi, *Rheum palmatum* L. and *Gardenia jasminoides* Ellis. The cure rate reaches as high as 90% and the mortality rate is kept to less than 10%. Epidemic encephalitis morbidity rate of 20–30%, and is treated with *Coptis teeta* Wall. and *Lonicera japonica* Thunb. Its cure rate for the disease is over 50–60% and the mortality rate is kept to less than 40%. For viral gastroenteritis and dysentery, the morbidity rate is as high as 65% and most cases occur when the pigs are young. It manifests as diarrhea and locals use a mixture of *Artemisia carvifolia* Buch.-Ham. ex Roxb., *Bidens pilosa* L., plant ash, and *Cornus kousa* F. Buerger ex Hance to make a decoction to treat this problem. The cure rate for the disease is as high as 95% and the mortality rate is kept to less than 5%. For illnesses caused by wind and cold, locals use *Pseudognaphalium affine* (D. Don) Anderberg, *Artemisia yunnanensis* J. F. Jeffrey ex Diels, *Equisetum ramosissimum* subsp. *debile* (Roxb. ex Vauch.) Hauke, and *Plantago depressa* Willd. to clear heat, detoxify the body, and reduce inflammation and fever in a decoction.

### 3.2. The Food Culture Derived from Gaoligongshan Pig

In the Nu River watershed, the Lisu, Bai, and Yi ethnic groups commonly use the Gaoligongshan pig as their main source of protein. They have blended their consumption with their characteristics to form a distinctive diet culture.

#### 3.2.1. The Shazhufan Dishes

The Gaoligongshan pigs are usually slaughtered in winter before the Spring Festival (the first day of January in Chinese lunar calendar is New Year. The pig slaughter is an important social event when friends and family members are invited to join in the slaughtering ceremony, cleaning, dish preparing, and tasting pork as everyone gathers to welcome the Spring Festival. Thus, “*Shazhufan*” is also known as “pre-*Chunjie* party”. Various *Shazhufan* dishes will be prepared for the party. Different parts from the pork are used to make *Shazhufan* dishes. Among them, “*Duo Sheng*”, the thinly sliced raw pork, is one of the representative dishes. This dish, together with other local cuisines, had formed the Nujiang Bai diet culture. In the Ming Dynasty (1368–1644 A.D.), the book *Yunnan Tujing Zhishu* recorded that “whenever a marriage (feast) is held, raw meat of all kinds is used, finely chopped, and eaten with garlic paste, and this is a precious thing”. The dish’s name is “E Y in He Xiao” in Bai language, which is generally translated as “eating raw skin” in Chinese. It is prepared by slicing the roasted pork, pork skin, and pork liver into thin slices and serving them with a prepared dipping sauce. This is a precious diet culture inherited from the ancestors of the Bai people who once lived on this land, and the Bai custom of eating raw meat continues nowadays.

#### 3.2.2. Huo Tang Culture

The Nujiang Prefecture has historically been far from transportation and commercial centers because of geographical isolation. Fortunately, the climate in Nujiang River watershed is relatively mild all the year. The ethnic minorities living in the watershed, such as the Lisu and Bai, have taken advantage of the environment to create a “*Huo Tang*” culture suitable for location. “*Huo Tang*”, also known as the “*Huo Keng*”, was used for cooking during the day and for warming the fire at night, with constant smoke throughout the year, and was an important place for heating, lighting, cooking, sleeping and for socializing, meeting and deliberating and worshipping the spirits. Almost every Nujiang Bai family has a *Huo Tang*, which has also become the seat of the ancestors. In local beliefs, the *Huo Tang* is the family god or family ghost that governs the whole family and has a beautiful symbolic meaning of warding off bad luck.

Pipa pork and Laowo ham are two local products directly associated with *Huo Tang* culture from the GPs (Figure 6). Both products have been recognized as the Intangible Cultural Heritage List in Yunnan. The Laowo Ham is produced in Laowo Town, Lushui City, Nujiang Prefecture, Yunnan Province [33]. There is a saying that in the first year of the Yongli reign of the Ming Dynasty (1646 AD), a descendant of the Dali clan visited Laowo and inadvertently tasted Laowo ham, which became an exclusive tribute to the nobility because of its delicious taste. The unique curing process of Laowo ham combined with the smoking method results in a nutritious and healthy delicacy with nitrite levels in the pork below national standards. Moreover, forming the microbiota is benefit to digestion and gut health. *Pipa* pork is also produced using a combination of curing and smoking. After curing, the meat is smoked for six months over *Huo Tang* using *Pinus yunnanensis* Franch. or *Alnus nepalensis* D. Don before the meat is ready. The smoked meat has the distinctive aroma of firewood and is served in a stew over the *Huo Tang.* The use of fire has been around for a long history. People from different places have creatively used different processes and smoking methods to cook food [34,35]. It not only enriches people’s dietary needs but also develops and preserves the precious food culture.

#### 3.2.3. Folk Food Therapy Culture

The local people have knowledge of the food therapy derived from Gaoligongshan pigs. For example, when children have enuresis, the elderly stew pig’s bladder with *Ajuga forrestii* Diels to treat this ailment. The pig’s hoof nails are collected and scorched, and ground into powder. The patient takes powder with warm water orally to relieve stomach problems. Pig’s bile is an anti-inflammatory medicine that clears heat and relieves cough. The pig’s gallstones are ground into powder to resolve phlegm.

### 3.3. The Sacrificial Culture of Gaoligongshan Pig

The Gaoligongshan pig has become an essential animal in annual festivals and significant events such as marriages, funerals, children’s full moon, building houses, and setting graves. It has been dedicated for use in praying for good weather and rain, good harvests, family peace, and prosperity for their livestock.

#### 3.3.1. The Sacrificial Culture of Bai

The Bai have unified ritual specifications and requirements for the placement of offerings, which are summarized as “incense, flowers, lamps, fruit, tea, food, treasure and clothes”. The sacrificial site is divided into “*Shang Tang*” (up space) and “*Xia Tang*” (down space). Those placed around the corners of the table and on the ground are called “*Xia Tang*”, and are dedicated to the Lord of the Land. The pig is the primary offering for the “*Shang Tang*.” On the right side of the *Dou* (a traditional woody container to measure rice) the pork head was laid, together with a bowl of pork offal. The skinned head of the pig is placed on the left side of the *Dou*. The *Dou* will be filled with rice. The “*Xia Tang*” is decorated with fruits. In addition, the “*Shang Tang*” and “*Xia Tang*” will be decorated with “*Liu Yan Liu Mi*”, that is, one jar of rice, one jar of tea, one jar of salt, and one bowl of beans. Three incense sticks are burned. Other offerings, including paper money, a pair of candles, an oil lamp, a cup of tea, liquor, and two bowls of rice, will be also provided in the traditional Bai rituals.

#### 3.3.2. Shangwang Festival of the Bai’s Lemo People

Lemo is a branch of Bai ethnic group in Nujiang Prefecture. The *Shangwang* Festival is an important festival for the Lemo people in Luobenzhuo, a township in Lushui. The festival has been listed in the Intangible Cultural Heritage Protection Project of Yunnan Province. “*Shangwang*” means March and is also known as the March Festival that is usually chosen every year in March, with dates between 1 March and 12 March. It is based on the first pig day of the 12 Chinese zodiac signs and marking the beginning of a farming year. The locals designate the first pig day of March on the lunar calendar as Spring Farming Day. The day before the festival, pigs are slaughtered, liquor is prepared, and rice cake is made to worship the gods of heaven, praying for a good harvest and happiness to be brought to the Lemo people.

#### 3.3.3. The Kuoshi Festival of Lisu People

The *Kuoshi* Festival is the largest traditional festival of the Lisu people in Nujiang and is a representative project of the Intangible Cultural Heritage of Yunnan Province. It has lasted for 500 years. During the *Kuoshi* Festival (20 December in the Chinese lunar calendar, usually lasting 3 days), every household kills a male pig. The head of the household puts the pork head, the front legand offal together in a dustpan and pours the “*Bu Zhi* liquor” over them to offer them to the ancestors (Figure 7). At this time, respected elders or priests take turns to visit each family to give blessings and recite a prayer of good family wishes. The hind legs of the pig are used to honor both parents. The rest of the pork will be preserved or smoked to become bacon, ham, and other products.

## 4. Conclusions

Our research has documented and summarized the rearing management culture of the Gaoligongshan pigs, including the traditional uses of forage plants, the semi-wild and free-range rearing pattern, and traditional Chinese veterinary practice for disease control. There are rich food culture, various festivals, and rituals culture associated with the Gaoligongshan pig of the ethnic minorities in the Nujiang region, mainly in the Bai and Lisu communities. This study also reveals the importance and irreplaceability of the semi-wild and free-range model in the rearing of Gaoligongshan pig. The holistic conservation strategy of traditional knowledge related to biocultural diversity for species conservation, community development, and cultural transmission is proposed, in terms of our investigations. The local people’s experiences from the investigation sites are significant to rural revitalization in the Nujiang River watershed. They can also serve as a reference for conserving and developing livestock and poultry genetic resources in other ethnic areas.

## Figures and Tables

**Figure 1 biology-11-01603-f001:**
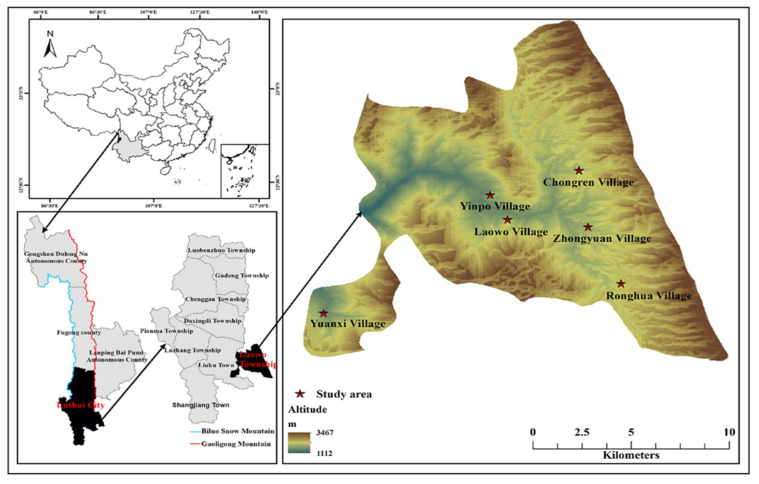
Map of the study area.

**Figure 2 biology-11-01603-f002:**
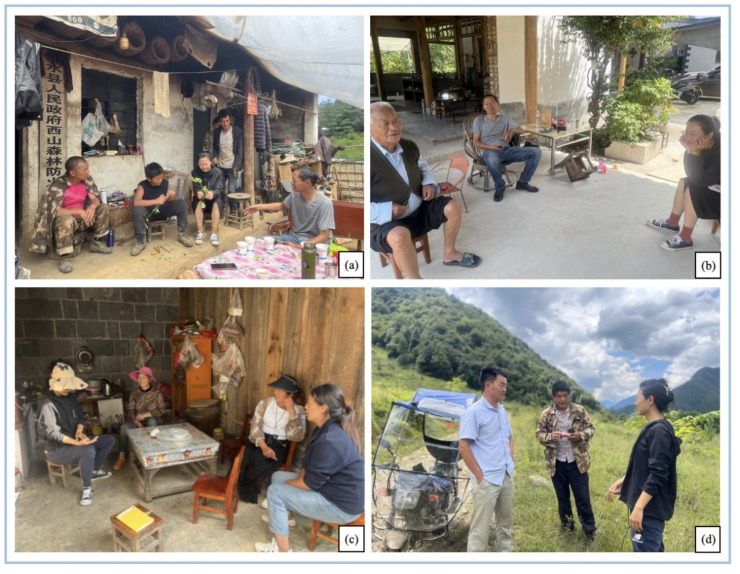
Ethnobiological surveys of Gaoligongshan pigs. (**a**–**c**) Semi-structured interviews in different households; (**d**) participatory observation in the field. (Photo by Xingcen Zhao, August 2022, Photographed separately in Laowo, Ronghua, Yuanxi and Chongren villages).

**Figure 3 biology-11-01603-f003:**
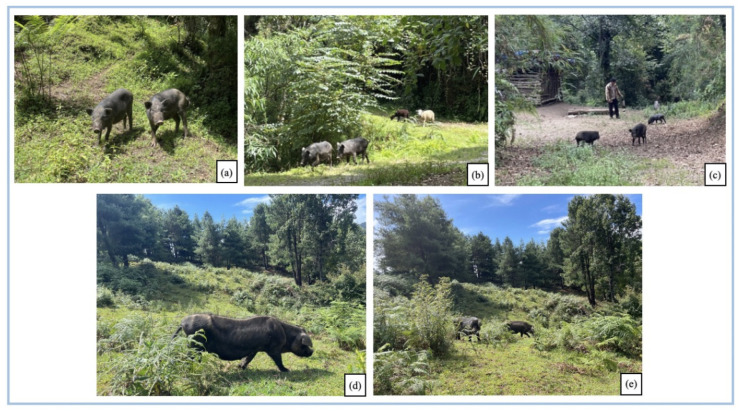
The semi-wild and free-range model to manage Gaoligongshan pigs. (**a**) Gaoligongshan pig cubs; (**b**) Gaoligongshan pigs mixed with cattle and sheep in grazing; (**c**) a farmer is feeding the piglets; (**d**) an adult Gaoligongshan pig; (**e**) the living habitat of Gaoligongshan pigs. ((**a**–**c**) photo by Xingcen Zhao, (**d**,**e**) photo by Yanan Chu, August 2022, (**a**–**c**) photographed in Laowo village, (**d**,**e**) photographed in Chongren village).

**Figure 4 biology-11-01603-f004:**
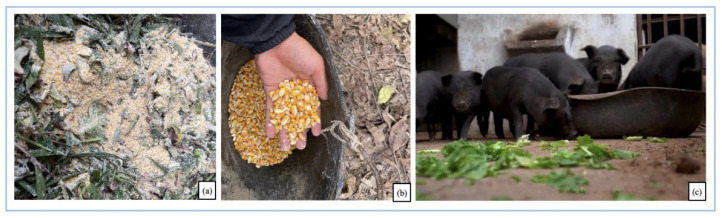
Commonly cultivated plants in artificial supplemented. (**a**) The maize flour mixed with wild forage, fed piglets around 4–5 months of age; (**b**) the corn kernels, fed piglets after 6 months of age; (**c**) the cultivated plants for daily feeding. (Photo by Yanan Chu, August 2022, (**a**) photographed in Yinpo Village, (**b**,**c**) photographed in Zhongyuan Village).

**Figure 5 biology-11-01603-f005:**
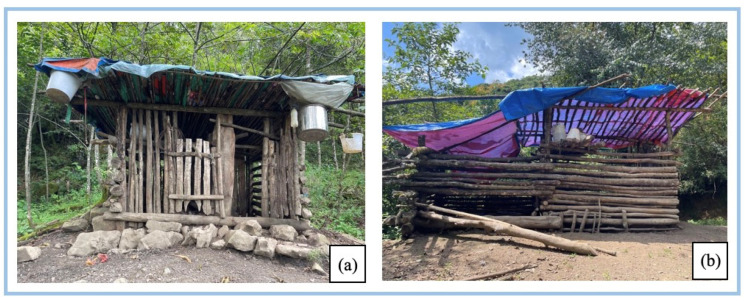
The pigsty built by farmers halfway up the hill. (**a**) The small-scale pigsty, which can hold up to 5 pigs; (**b**) the slightly large-scale pigsty, which can hold about 10 pigs. (Photo by Xingcen Zhao, August 2022, Photographed separately in Chongren and Zhongyuan villages).

**Figure 6 biology-11-01603-f006:**
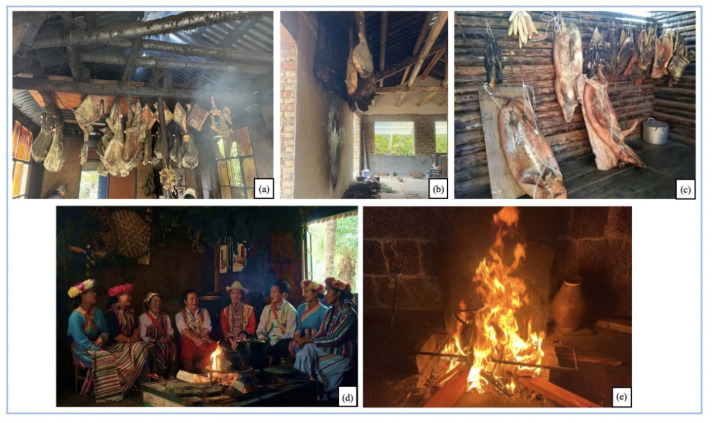
Ham products preserved by different farmers. (**a**,**b**) The preserved ham hung in the farmer’s house kitchen (also dining room) over the *Huo Tang*; (**c**) the fermented Pipa pork stored in a cool, ventilated, and dry place, neat the *Huo Tang*; (**d**) the ethnic minority people sitting around the *Huo Tang* chatting; (**e**) the burning fire *Huo Tang*. ((**a**,**b,e**) photo by Yanan Chu, August 2022, Photographed in Laowo Village; (**c**,**d**) photo by Nujiang Lisu Autonomous Prefecture Culture and Tourism Bureau, July 2020, Photographed in Bingzhongluo Village).

**Figure 7 biology-11-01603-f007:**
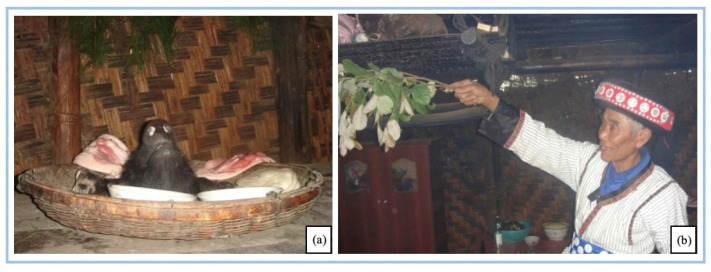
The Lisu people worship their ancestors at the *Kuoshi* Festival. (**a**) Ritual objects; (**b**) village priest clearing houses with *Lithocarpus* branches while blessing. (Photo by Nujiang Lisu Autonomous Prefecture Culture and Tourism Bureau, July 2020, Photographed in Zhongyuan Village).

**Table 1 biology-11-01603-t001:** Questions used for semi-structured interviews.

No.	Question
1	Why do you choose to breed Gaoligongshan pigs?
2	How long have you practiced Gaoligongshan pig farming?
3	What is the daily management of the Gaoligongshan pig?
4	Do you know what wild plants the Gaoligongshan pig likes to eat most?
5	The wild plants that Gaoligongshan pigs like to eat will change as they grow?
6	Will the mix of supplementary feeding materials be adjusted according to the growth stage or season?
7	Where is the pigsty built and why did you select there?
8	What are the common diseases of the Gaoligongshan pig?
9	How does the application of Chinese veterinary medicine to Gaoligongshan pigs and the treatment work?
10	What are the local traditional cultures related to the Gaoligongshan pig?
11	What are the processing methods for Gaoligongshan pig?
12	What are the sacrificial rituals of the different ethnic minority festivals?
13	Do your children actively participate in these traditional-culture-related activities?

**Table 2 biology-11-01603-t002:** Ethnobiological inventory of forage plants to feed Gaoligongshan pigs and other traditional uses.

Voucher No.	Family	Scientific Name	Local Name	Parts as Forage	Habit	Consumed Time	Preference	Abundance	Medicinal Use for Human Ailments	Human Food and Other Uses
LH089	Amaranthaceae	*Amaranthus retroflexus* L.	Da gu cai	Leaves	Herb	except winter	**	††	Root can treat dermatitis in line with the disease	The stems and leaves can be eaten when cooked
LH098	Amaranthaceae	*Chenopodium album* L.	Hui tiao cai	Aerial part	Herb	except winter	*****	††††	The whole herb can stop diarrhea and itching	The stems and leaves can be eaten when cooked, spices
LH132	Apiaceae	*Oenanthe javanica* (Bl.) DC.	Shui qin cai	Leaves	Herb	All available time	*****	†††††	Lowering blood pressure	The stems and leaves can be eaten when cooked
LL042	Apiaceae	*Centella asiatica* (L.) Urban	Ma ti cai	Aerial part	Liana	All available time	*****	††††	The whole herb can reduce fever and causing diuresis, subduing swelling and detoxicating, injuries from falls, fractures, contusions and strains, and traumatic injury	Stems and leaves raw and cooked can be eaten
LH025	Caryophyllaceae	*Drymaria cordata* (Linnaeus) Willdenow ex Schultes	Yue liang cao	Aerial part	Herb	All available time	*****	††††	The whole herb can diminish inflammation, heat clearing-detoxication	/
LH138	Caryophyllaceae	*Stellaria media* (L.) Villars	Ji chang cai	Aerial part	Herb	Exept winter	*****	††††	Cauline leaf can promote blood circulation and remove blood stasis/ heat clearing/detoxication	/
LH017	Caryophyllaceae	*Stellaria aquatica * (L.) Scop.	E chang cai	Aerial part	Herb	Exept winter	****	††††	/	/
LH071	Commelinaceae	*Commelina communis* L.	Ya zhi cao	Aerial part	Herb	Exept winter	**	†††	Diuresis and making tumor disappear, heat clearing/detoxication, diminish inflammation	/
LH140	Compositae	*Cirsium chlorolepis* L.	Ji chi gen	Leaves	Herb	All available time	***	†††		The roots can be eaten when cooked
LH093	Compositae	*Taraxacum * *mongolicum * Hand.-Mazz.	Pu gong ying	Aerial part	Herb	Exept winter	***	†††††	Heat clearing-detoxication, clearing liver and improving vision, removing swelling and lumps, dispelling wind and removing obstruction in the meridians	Aerial part raw and cooked can be eaten
LH058	Compositae	*Bidens pilosa* L.	Yang cha cao	Aerial part	Herb	Exept winter	**	††††	Heat clearing/detoxication, diminish inflammation, activate blood flow and remove blood stasis, and treat snake bites, injuries from falls, fractures, contusions and strains, traumatic injury	The stems and leaves can be eaten when cooked
LH005	Compositae	*Cirsium japonicum * Fisch. ex DC.	Ji chi gen	Aerial part	Herb	All available time	*	†††	Enrich the calcium	The roots are stewed with pork
LH096	Compositae	*Sonchus oleraceus* L.	/	Aerial part	Herb	All available time	***	†††	The whole herb can clear damp; heat clearing/detoxication	The stems and leaves can be eaten when cooked
LT076	Moraceae	*Broussonetia papyrifera* (L.) L’Heritier ex Ventenat	Gou shu	Leaves	Tree	All available time	*****	†	Herb can eliminate heat, diuresis and make tumors disappear; detoxification	Flowers can be steamed or cooked cold, stem can make paper and craft products
LL119	Oxalidaceae	*Oxalis corniculata* L.	Suan mu gua/ Mu gua cao	Aerial part	Liana	Exept winter	*****	††††	The whole herb can eliminate heat, diuresis and make tumors disappear, reinforcing the spleen to promote digestion and prevent diarrhea	Aerial parts can be eaten raw
LH014	Plantaginaceae	*Plantago depressa * Willd.	Lai ha ma ye	Aerial part	Herb	All available time	****	†††††	The whole herb can heat clearing-detoxication, diminish inflammation, diuresis	Aerial part raw and cooked can be eaten
LH003	Polygonaceae	*Persicaria runcinata * (Buch.-Ham. ex D. Don) H. Gross	Ha zhu cao	Aerial part	Herb	All available time	*****	†††††	/	/
LH007	Polygonaceae	*Persicaria chinensis * (L.) H. Gross	La liu cao	Aerial part	Herb	All available time	*****	†††††	/	/
LH055	Polygonaceae	*Fagopyrum dibotrys * (D. Don) Hara	Suan jiang cao	Aerial part	Herb	Exept winter	*****	††††	The tuberous root can heat clearing-detoxication, remove blood stasis and expelling pus, stimulate appetite and digestion, relax the bowels	The stems and leaves can be eaten raw or cooked
LH097	Polygonaceae	*Rumex acetosa* L.	Yang pi jin	Aerial part	Herb	Exept winter	*****	††††	The whole herb can be used for clearing heat and detoxicating	/
LH130	Polygonaceae	*Persicaria Senticosa * var. *sagittifolia * (H. Lév. et Vaniot) Yonekura et H. Ohashi	Ha zhu cao	Aerial part	Herb	Exept winter	*****	†††††	/	/
LH020	Polygonaceae	*Persicaria microcephala * (D. Don) H. Gross	Ha zhu cao	Aerial part	Herb	Exept winter	*****	††††	/	/
LS019	Rosaceae	*Rubus multisetosus * Yü et Lu	La la guo	Fruits	Shrub	All available time	**	††††	Relieve a cough, to stop pain, neutralize the effect of alcoholic drinks	/

Species in this inventory are ordered by the family name alphabetically. The local name of forage plants is written in Chinese *Pinyin*. All the tables below follow this format. * in preference represents the preference level of Gaoligongshan pig (GP). The higher the number of *, the more common the corresponding plant is in the diet of Gaoligongshan pig; † in abundance represents resource amount of the forage plants. The higher the number of †, the more widely the corresponding plant is distributed in the study area.

**Table 3 biology-11-01603-t003:** Inventory of cultivated plants as Gaoligongshan pigs’ forage and as human food.

Voucher No.	Family	Scientific Name	Local Name	Parts as Forage	Life Form	Consumed Time	Preference	Feed Proportion	For Human Consumption
LH125	Amaranthaceae	*Amaranthus caudatus* L.	Su mi cai	Aerial part	Herb	Exept winter	**	**	Stems and leaves young and tender
LH080	Araceae	*Amorphophallus konjac* C. Koch	Shui guo cai	Leaves	Herb	Exept winter	***	**	Tubers
LH100	Brassicaceae	*Brassica pekinensis* (Lour.) Rupr.	Da bai cai	Aerial part	Herb	All available time	*****	*	Aerial part
LH079	Cannaceae	*Canna edulis* Ker	Ye ba jiao	Aerial part	Herb	Exept winter	*****	**	Leaf core
LH103	Compositae	*Lactuca sativa* var. *angustata* Irish ex Bremer	Wo ju	Leaves	Herb	All available time	*****	*	Stems
LH105	Convolvulaceae	*Ipomoea aquatica* Forsskal	Kong xin cai	Aerial part	Herb	Exept winter	*****	*	Stems and leaves young and tender
LH107	Convolvulaceae	*Ipomoea batatas* (L.) Lamarck	Shan yao	Leaves and rootstock	Herb	Exept winter	*****	**	Tubers
LH057	Cucurbitaceae	*Cucurbita moschata * (Duch. ex Lam.) Duch. ex Poiret	Gua ye	Aerial part	Herb	Exept winter	*****	*	Stem apex and fruit
LH081	Cucurbitaceae	*Sechium edule* (Jacq.) Swartz	Yang si gua	Aerial part	Herb	Exept winter	*****	*	Stem apex and fruit
LH099	Fabaceae	*Phaseolus vulgaris* L.	Dou ye	Leaves	Herb	Exept winter	****	**	Fruit
LH092	Poaceae	*Zea mays* L.	Bao gu	Maize stalk and fruits	Herb	All available time	*****	****	Fruit

* in preference represents the preference level of Gaoligongshan pig. The higher the number of *, the more common the corresponding plant is in the diet of Gaoligongshan pig. The * in feed proportion represents the number of seasons in which corresponding plants appear as feed proportion.

**Table 4 biology-11-01603-t004:** Common plants for keeping pigsty dry.

Voucher No.	Family	Scientific Name	Local Name	Parts Consumed	Habit	Consumed Time	Abundance
LT039	Pinaceae	*Pinus yunnanensis* Franch.	Jia song	Pine needles	Tree	All available time	***
LH126	Poaceae	*Setaria plicata* (Lam.) T. Cooke	/	Aerial part	Herb	All available time	****
LH112	Poaceae	*Setaria viridis* (L.) Beauv.	Gou wei ba cao	Aerial part	Herb	All available time	****
LH045	Thelypteridaceae	*Pteridium aquilinum* (L.) Kuhn var. *latiusculum* (Desv.) Underw. ex Heller	Jue cai	Aerial part	Herb	Spring and Summer	**

* in abundance represents resource amount of the forage plants. The higher the number of *, the more widely the corresponding plant is distributed in the study area.

**Table 5 biology-11-01603-t005:** Selected medicinal plants used for treating Gaoligongshan pigs’ common diseases.

Voucher No.	Family	Scientific Name	Local Name	Parts Consumed	Life Form	Consumed Time	Medicinal Efficacy for Pig	Method of Uses	Attending Disease
LS156	Caprifoliaceae	*Lonicera japonica* Thunb.	Jin yin hua	Flowers	Shrub	Summer	Clearing heat and detoxicating, diminish inflammation and detumescence	Decoct medicinal herbs	Virus encephalitis
LH157	Compositae	*Pseudognaphalium affine* (D. Don) Anderberg	Huang hua	The whole plant	Herb	Except the winter	Antitussive and expectorant effects, bring down a fever, wound healing	Decoct medicinal herbs	Diseases caused by cold factors
LH158	Compositae	*Artemisia* *yunnanensis* J. F. Jeffrey ex Diels	Ye ba hao	Aerial part	Herb	All available time	Clearing heat and detoxicating, bring down a fever	Decoct medicinal herbs	Diseases caused by cold factors
LH058	Compositae	*Bidens pilosa* L.	Yang cha cao	The whole plant	Herb	Except the winter	Clearing heat and detoxicating, promoting blood circulation to remove blood stasis, antidiarrheal, reducing inflammation, venomous snake bite, traumatic injury	Decoct medicinal herbs	Diarrhoea
LH159	Compositae	*Artemisia carvifolia* Buch.-Ham. ex Roxb.	Hao zi	The whole plant	Herb	All available time	Clearing heat and detoxicating, relieving summer-heat, dispelling wind and arresting itching	Decoct medicinal herbs	Diarrhoea
LT160	Cornaceae	*Cornus kousa* F. Buerger ex Hance	Huo shan li zhi	Flowers and leaves	Tree	All available time	Antidiarrheal	Decoct medicinal herbs	Diarrhoea
LH108	Equisetaceae	*Equisetum ramosissimum* subsp. *debile* (Roxb. ex Vauch.) Hauke	Jie jie cao	The whole plant	Herb	All available time	Clearing heat and detoxicating, reduce inflammation, stop pain and relieve itching	Decoct medicinal herbs	Diseases caused by cold factors
LH164	Lamiaceae	*Scutellaria baicalensis* Georgi	/	Rootstock	Herb	All available time	Heat-clearing and damp-drying drug, purging intense heat and detonicating, hemostasis, antidiarrheal, miscarriage prevention	Decoct medicinal herbs	Vesicular exanthema
LH014	Plantaginaceae	*Plantago depressa* Willd.	Lai ha ma ye	The whole plant	Herb	Except the winter	Clearing heat and detoxicating, diminish inflammation and diuresis	Decoct medicinal herbs	Virus nephritis
LH161	Polygonaceae	*Rheum palmatum* L.	Da huang	Roots	Herb	All available time	Purging heat and bowels, clearing heat and detoxicating, hemostasis	Decoct medicinal herbs	Vesicular exanthema
LH162	Ranunculaceae	*Coptis teeta* Wall.	/	Roots	Herb	All available time	Clearing heat and detoxicating, diminish inflammation and detumescence	Decoct medicinal herbs	Virus encephalitis and vesicular exanthema
LS165	Rubiaceae	*Gardenia jasminoides* Ellis	/	Fruits	Shrub	Autumn	Reducing fever and causing diuresis, cool the blood and eliminating stasis, detoxicate	Decoct medicinal herbs	Vesicular exanthema
LT166	Rutaceae	*Phellodendron chinense* Schneid.	Huang bai	Bark	Tree	All available time	Clearing hectic heat, purging intense heat and detonicating, antibacterial	Decoct medicinal herbs	Vesicular exanthema

## Data Availability

All data are included in the manuscript.

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
