# Peer review of "A Biocultural Study on Gaoligongshan Pig (Sus scrofa domesticus), an Important Hog Landrace, in Nujiang Prefecture of China"

_biology, 2022, doi:10.3390/biology11111603_

Round 1

Reviewer 1 Report

The topic is interesting and up-to-date and suitable for publication in this special issue. However, before publication, the reviewer has questions and comments to which I would like to be answered.

The Latin name is incorrect in the Title and Abstract (line 28) „Susscrofa domestica“. Should be „Sus scrofa domesticus“. https://www.ncbi.nlm.nih.gov/Taxonomy/Browser/wwwtax.cgi?id=9825

I suggest changing the title because it's bit unintelligible right now („Biocultural Study on Gaoligongshan Pig (Susscrofa domestica), An Important Hog Landrace in the Eastern Himalayas“). E.g: „A biocultural study of an important hog landrace, the Gaoligong Mountain pig (Sus scrofa domesticus), in the Eastern Himalayas, China.“, or something similar but the country where the study was done (China), should definitely be reflected in the title.

Question: How did you get the English name of this native pig? According to the place name, it should be "Gaoligong pig". "Shan" ( å±± )translates to "mountain" in English and should have been reflected in the name. The name of the pig is on China's list of endangered breeds is 高黎贡山猪 („Gaoligong Mountain Pig“) http://www.nahs.org.cn/zt_10027/xqpc/202104/P020210406591174266284.pdf (page 68). "Gaoligong Mountain Pig" would be correct when translated into English. Since this breed was named by scientists. The question, how the local people call this breed of pig? How did they differentiate it from other pig breeds at the name level? After all, Yang et al (2011) mentions that in this area are quite a lot of local breeds of pigs (you should also refer this article becouse this analyzes on the history of domestication of domestic pigs in this region).

Yang, S., Zhang, H., Mao, H., Yan, D., Lu, S., Lian, L., ... & Gou, X. (2011). The local origin of the Tibetan pig and additional insights into the origin of Asian pigs. PloS one6(12), e28215.

You also was had first question "What breed of pigs do you keep?", which shows that more pig breeds are keeping in the region. How many different pigs breeds were named to you by the local people?

Regarding the importance of the topic, you might want to read and refer to articles published also outside of China (currently mostly Chinese researchers are in the references), e.g.:

Molnár, Z., Szabados, K., Kiš, A., Marinkov, J., Demeter, L., Biró, M., ... & Babai, D. (2021). Preserving for the future the—once widespread but now vanishing—knowledge on traditional pig grazing in forests and marshes (Sava-Bosut floodplain, Serbia). Journal of Ethnobiology and Ethnomedicine17(1), 1-30.

You should also mention that today there is pig breeding in the area in large farms. Read (and you may want to refer) here: („Gaoligong Mountain Pig Conservation and Breeding Farm has become a breeding technology learning base for Farmers in Nujiang“) https://www.laitimes.com/en/article/c8j7_cbbc.html Or watch a movie https://www.youtube.com/watch?v=_P7y3D0U6yo

In line 113 you say "breeding" and in line 117 "cultivated". Change "cultivated" to "breeded".

In lines 122-124: „Informants including 21 female and 68 male, with an average age of 51.6 years, from 10 to 84 years old, have practiced pig farming for an average duration of 24.5 years.“. I would suggest changing the end of this sentence. It is better to say that: „Informants including 21 female and 68 male, with an average age of 51.6 years, from 10 to 84 years old, who have come into contact with this pig farming during their lifetime.“. Your youngest 10-year-old respondent is still a child. He was talking about his parents' pig farming, not his own pig farming. Therefore, bringing out the average is not reasonable here. After all, you didn't ask how long people have practiced pig farming (see the questions in Table 1).

Figures 4–7 are missing the authors of the photos and the date of taking the photos. It is also missing in which villages pictures was taken. Since you only have people's verbal consent, you can't add village accuracy to photos with people on them.

Shicai et al (2010), who have done an ethnoveterinary study on the "Gaoligongshan pig" in the same area, say that this pig breed has only recently been discovered (described). There is no information in your manuscript, who first described this pig breed and when? Also, you have not cited the Shicai et al (2010) paper and did not use it in your analysis (eg for comparative analysis).

Shicai, S., Andreas, W., & Vernooy, R. (2010). The importance of ethnoveterinary treatments for pig illnesses in poor, ethnic minority communities: a case study of Nu people in Yunnan, China. International Journal of Applied Research in Veterinary Medicine8(1), 53-59.

You list four studies that have examined the quality of pork. However, there are more of these studies, e.g. you may want to refer this also:

Zhao Guiying, Duan Bofang, Duan Xingquan & Ji Xiaorui, 2012. Comparison of Meat Quality and Composition for Longissimus Muscle Tissues from Gaoligongshan Pig and Saba x Gaoligongshan Cross Pig. Journal of Animal and Veterinary Advances, 11: 592-594.

In the chapter "Study area", you have not indicated that it is a Gaoligong Mountain Biosphere Reserve https://en.unesco.org/biosphere/aspac/gaoligong So you should mention what are the objectives of this biosphere reserve. It is also not possible to read the county and region names correctly from Figure 1 (too small letters). You should write the names of the counties on the map in larger size. Also highlight the names of the mountains mentioned in the text on the colored map (map with villages).

Review your references and align them with the journal's requirements (https://www.mdpi.com/journal/biology/instructions ). Currently, they do not meet the requirements of the journal. For example, the names of cited journals must be abbreviated and italicized, etc. Also, remove the duplicating abbreviation (DOI:) “DOI: https://doi.org/” from references. Line 33 you have indicated reference [41] there is no such in the references. Please correct this reference.

In Tables 2, 3, 4, 5, you should mention "Local name" wheather a plant names said by members of the Bai, Lisu, or Han linguistic group. You also have to indicate in "Data collection" (lines 121-122) how many members of the Bai, Lisu, or Han ethnic language groups were among the 79 respondents.

The numbers 1 to 10 are written out, not given as a number. See for example line 226 "more than 5 pigs." should be "more than five pigs."

Line 332, ad what kind of pine species is it? Pinus yunnanensis? Alnus sp. (A. incana and A. glutinosa) wood and pine (Pinus sylvestris) cones are also used for smoking meat in Europe, you could also mention this e.g. in Estonia:

Kalle, R., & Sõukand, R. (2012). Historical ethnobotanical review of wild edible plants of Estonia (1770s-1960s). Acta Societatis Botanicorum Poloniae81(4).

Kalle, R., & Sõukand, R. (2016). Current and remembered past uses of wild food plants in Saaremaa, Estonia: Changes in the context of unlearning debt. Economic Botany70(3), 235-253.

The question is whether pine wood or pine cones were used to smoke the meat? Pine has a lot of tar and therefore produces a lot of black, bitter-tasting soot on the smoked meat. That is why, for example, pine wood is not used in Estonia, only dry cones that have fallen from the tree.

Since the article's readers are international audiences who are not familiar with the Chinese calendar, you should specify the dates (or range of dates) when local festivals take place: Spring Festival (line 294), Shangwang Festival (line 367), and Kuoshi Festival (line 376).

You say "threatened", "danger", "endangered" only in the Introduction (line 68-77; 86-89). But in the text you do not point out what are the threats to this pig breed today? Crossbreeding with other pig breeds? Disappearance of local traditions? New diseases such as African swine fever? Change and loss of natural habitats? So you have to bring out what are the major threats to the survival of this breed today. Also, in the results and discussion, you should more clearly outline how your work helps to overcome the given threats, so that the given pig breed is preserved?

Author Response

Thank you very much  for your kind comments and suggestions. We have revised our manuscript according to your suggestions. The point-by-point response was attached.

Reviewer 2 Report

Dear authors,

After carefully reading your manuscript, I have a few point-by-point comments and remarks, please see them bellow:

Title: The correct Latin name of the studied species is ‘Sus scrofa domestica’, please correct this here and throughout the manuscript;

Semi-structured interviews sound without a research approach in mind, rather than interviews with open questions if that was the case; The questionnaire presented by the authors has only 9 questions (table 1) and they are very general, this is a weak point of the study;

There is a lack of information and data in the abstract section, please include the most important aspects of your findings;

Lines 52-53: ‘It has been proven that genetic resources cannot be conserved in isolation from their original natural and cultural environment.’ I find this statement as false, since a great number of ex situ techniques exist or are developed for the conservation of various farm animal genetic resources. Please rephrase or delete the sentence;

Lines 80-90: When describing the Gaoligongshan pig, please add some much-needed information such as census (estimate on the number of pigs left), average littler size, growth rates and maybe some health resistance, since you mention this factors as the main downfall reasons for which the breed has been abandoned;

L 15: You mention that the Gaoligongshan pigs are a rare and an endangered breed, however, in Line 11 you state ‘…are massively raised…’, this is conflicting, maybe replace with primarily raised/found;

L 117: Pigs are not cultivated, are being reared or raised/found;

L 136: I do not think that it is necessary to nominate the names of who identified the plat species at this stage, since they are mentioned at the end of the manuscript;

L 239: Please rephrase ‘pig usually starts to have sex at around four…’, with something on the lines of ‘puberty in GP pigs is reached at around 4 months of age’;

L 240: I honestly doubt that the oestrous cycle and gestation length are influenced at a large extent by the environment, the influence could be of no more than 2 or 3 days, other factors play a more significant role, please rephrase;

L241-242: How can the authors state ‘estrogen secreted during oestrus affects the sow's metabolism’? When no such studies were made in the manuscript to back up their research;

L 243: ‘which attracts wild boars within a range of 3-5 242 km and allows mating to take place’, takes me to the conclusion that GP is not an actual breed, rather a crossbreed between local pigs and wild boars, please clarify this, since you state the name of the so called breed, even in the title and this is the main objective of the study, as to characterize the breed in its natural areal;

Do farmers keep GP boars? From the manuscript reads that the females only breed with wild boars, which through genetic absorption, would turn the GP into a wild boar population in a matter of 2-3 generations. If they use GP boars, what is the sex ratio?

Subsection 3.1.3.: What is the prevalence of this diseases, what are the morbidity and mortality rates? What is the survival rate of each disease? No data on this aspect was presented, which is a main flaw of the manuscript.

After reading this manuscript, although it has an interesting approach, I am not sure that is it in the aims of Biology journal, since the topic is more anecdotal and has a primarily social component, rather than science based. When it comes to data collection and presentation, the studied pig breed is only mentioned, however, the focus was on plant species found in the area.

Author Response

(The authors gave the same response as above.)

Round 2

Reviewer 1 Report

The authors have made the necessary corrections in the manuscript. I wish the authors success in publishing an interesting article.